# A Novel Signature of CCNF-Associated E3 Ligases Collaborate and Counter Each Other in Breast Cancer

**DOI:** 10.3390/cancers13122873

**Published:** 2021-06-08

**Authors:** Shu-Chun Chang, Chin-Sheng Hung, Bo-Xiang Zhang, Tsung-Han Hsieh, Wayne Hsu, Jeak Ling Ding

**Affiliations:** 1The Ph.D. Program for Translational Medicine, College for Medical Science and Technology, Taipei Medical University, Taipei 110, Taiwan; b101106043@tmu.edu.tw; 2International Ph.D. Program for Translational Science, College of Medical Science and Technology, Taipei Medical University, Taipei 110, Taiwan; 3Division of General Surgery, Department of Surgery, Taipei Medical University Hospital, Taipei 110, Taiwan; hungcs@tmu.edu.tw; 4Division of General Surgery, Department of Surgery, Taipei Medical University-Shuang Ho Hospital, Ministry of Health and Welfare, New Taipei City 23561, Taiwan; 5Department of Surgery, School of Medicine, College of Medicine, Taipei Medical University, Taipei 11031, Taiwan; 6Joint Biobank, Office of Human Research, Taipei Medical University, Taipei 110, Taiwan; thhsieh@tmu.edu.tw; 7Department of Biological Sciences, National University of Singapore, Singapore 117543, Singapore

**Keywords:** CCNF, FBXL8, FZR1, RRM2, E3 ubiquitin ligase, breast cancer, metastasis

## Abstract

**Simple Summary:**

The dysregulation of UPS exacerbates the tumor microenvironment and drives malignant transformation. As the largest family of E3 ligases, the SCF^F-boxes^ promotes BRCA progression. FBXL8 was recently identified to be a novel SCF E3 ligase that potently promotes BRCA. Here, we profiled the transcriptome of BRCA patient tissues by global NGS RNA-Seq and TCGA database analyses. A signature of four SCF^F-box^ E3 ligases (FBXL8, FBXO43, FBXO15, CCNF) was found to be pivotal for BRCA advancement. Knockdown of FBXL8 and FBXO43 reduced cancer cell viability and proliferation, suggesting their pro-tumorigenic roles. However, the overexpression of CCNF inhibited cancer cell progression, indicating its anti-tumorigenic role. FBXL8 and FZR1 pulled down CCNF, and double knockdown of FBXL8 and FZR1 caused CCNF accumulation. Additionally, CCNF partnered with a pro-tumorigenic factor, RRM2, and overexpression of CCNF reduced RRM2. Our findings suggest a potential for drugging CCNF in co-modulatory partnership with FBXL8 and FZR1, for anti-BRCA therapy.

**Abstract:**

Breast cancer (BRCA) malignancy causes major fatalities amongst women worldwide. SCF (Skp1-cullin-F-box proteins) E3 ubiquitin ligases are the most well-known members of the ubiquitination–proteasome system (UPS), which promotes cancer initiation and progression. Recently, we demonstrated that FBXL8, a novel F-box protein (SCF^F-boxes^) of SCF E3 ligase, accelerates BRCA advancement and metastasis. Since SCF^F-boxes^ is a key component of E3 ligases, we hypothesized that other SCF^F-boxes^ besides FBXL8 probably collaborate in regulating breast carcinogenesis. In this study, we retrospectively profiled the transcriptome of BRCA tissues and found a notable upregulation of four SCF^F-box^ E3 ligases (FBXL8, FBXO43, FBXO15, and CCNF) in the carcinoma tissues. Similar to FBXL8, the knockdown of FBXO43 reduced cancer cell viability and proliferation, suggesting its pro-tumorigenic role. The overexpression of CCNF inhibited cancer cell progression, indicating its anti-tumorigenic role. Unexpectedly, CCNF protein was markedly downregulated in BRCA tissues, although its mRNA level was high. We showed that both E3 ligases, FBXL8 and FZR1, pulled down CCNF. Double knockdown of FBXL8 and FZR1 caused CCNF accumulation. On the other hand, CCNF itself pulled down a tumorigenic factor, RRM2, and CCNF overexpression reduced RRM2. Altogether, we propose a signature network of E3 ligases that collaboratively modulates CCNF anti-cancer activity. There is potential to target BRCA through modulation of the partnership axes of (i) CCNF-FBXL8, (ii) CCNF-FZR1, and (iii) CCNF-RRM2, particularly, via CCNF overexpression and activation and FBXL8/FZR1 suppression.

## 1. Introduction

Breast cancer (BRCA) is amongst the top cancer killers worldwide. The ubiquitination–proteasome system (UPS) is involved in chronic inflammation-mediated tumorigenesis such as BRCA [1,2,3]. However, the underlying mechanisms and key factors through which UPS component drives BRCA metastasis are not fully understood. Characterizing global profiles of UPS factors followed by identifying specific UPS components and elucidating their molecular mechanisms of action in BRCA cells would be pivotal for the early diagnosis and treatment of the disease. The major enzymatic components of UPS include E1, E2, and E3. Ubiquitin is activated by E1 (ubiquitin-activating enzyme), which is conjugated to target proteins by E2 (ubiquitin-conjugating enzyme) and E3 (ubiquitin ligase) activities. The human genome encodes at least two ubiquitin E1s, fifty-three E2s, and approximately five hundred E3 enzymes [4]. Although some E2s can directly transfer ubiquitin to substrate proteins, in most ubiquitination processes, substrate selection and ubiquitin linkage are achieved by E3 [5,6]. 

Belonging to the largest family of the ubiquitin ligases is the multimeric SCF E3 ubiquitin ligase, which is composed of SKP1, a cullin, an F-box protein (SCF^F-boxes^), and RBX1 or RBX2. Importantly, the F-box factor is responsible for the recognition of target proteins for proteasomal degradation. Due to their ability to regulate the expression and activity of oncogenes and tumor-suppressor genes, SCF^F-boxes^ themselves play important roles in cancer development and progression [7,8]. Emerging experimental and clinical data indicate that SCF^F-boxes^ modulate cell cycle mediators, hence playing either tumor suppression or oncogenic function, and are tightly linked to cancer initiation and progression [9]. SCF^F-boxes^ are now recognized as potent targets of anticancer therapies. In addition to E3 ligases, different forms of E2–E3 pairing also determine the specificity of target recognition [10]. Although proteasome inhibitors such as Bortezomib have been approved by the FDA for cancer treatment, it was found to be cytotoxic, as it is a general inhibitor of protein degradation [11]. Thus, anti-cancer drugs targeting specific E3 ligases (or E2–E3 combinatorial cocktails) would be desirable for selective protein degradation aiming at the particular enzyme substrate(s). In this regard, evidence is accumulating to suggest undiscovered SCF^F-boxes^, which may be potentially involved in BRCA progression. Firstly, we demonstrated that FBXL8 (F-Box/Leucine Rich Repeat Protein 8) is a novel SCF^F-box^, which accelerates BRCA disease advancement and metastasis. Knockdown of FBXL8 caused the accumulation of two cancer suppressors (CCND2, IRF5), which inhibited BRCA progression [12]. Secondly, CCNF (a Cyclin F, also referred to as SCF^CCNF^), which is a component of an E3 ubiquitin–protein ligase complex, is reported to be tightly controlled by UPS [13]. However, CCNF-associated mechanisms are not fully clarified in BRCA, although its binding partner RRM2 (an oncogenic factor) was found to be overexpressed in numerous cancers, including BRCA [14]. Hence, we performed global profiling of UPS in primary clinical samples, particularly focusing on the SCF^F-boxes^ to study the profile of CCNF and identify its partners involved in BRCA progression.

Amongst the global profiles of UPS components, which were uniquely expressed in BRCA tissues, we found the upregulation of the mRNA levels of SCF^F-boxes^: FBXL8, CCNF, FBXO43, and FBXO15, which was affirmed by the large-scale analysis of the TCGA database. IHC staining consistently highlighted FBXO43, FBXO15, and FBXL8 [12]. Interestingly, the F-box CCNF (known as a cancer suppressor in hepatocellular carcinoma [13]) followed an opposite trend; although its mRNA level was upregulated, its protein level was downregulated. Furthermore, CCNF interacts with cognate E3 ligases, FBXL8 and FZR1, while it also promiscuously partners with a pro-tumorigenic factor, RRM2, and the overexpression of CCNF reduced RRM2. We propose a signature of CCNF-associated ubiquitin machinery that regulates BRCA development. 

## 2. Results 

### 2.1. Several UPS Components Are Significantly Upregulated in Primary BRCA Tissues 

We recently reported that FBXL8 is a novel F-box E3 ligase, which acts as a BRCA promoter, playing pivotal roles in anti-apoptosis [12]. Here, we further explored the clinical significance of the key components of the ubiquitin machinery in response to cancer progression by characterizing the RNA-Seq global profile to reveal the BRCA signature of UPS components. Figure 1A shows the experimental plan and strategic workflow in the current work. In particular, we focused on the E1, E2, and F-box genes, which are critical members of the SCF multimeric complex responsible for the specificity of targeted substrate recognition in BRCA patient tissues. Appendix A summarizes the clinicopathological parameters of the BRCA patients, which were also utilized for FBXL8 investigation [12]. Five breast carcinoma tissues and the corresponding normal counterparts were subjected to global mRNA profiling. Pathology information was provided by the medical records of BioBank, Taipei Medical University. The mean age of the BRCA patients was 59 years (range, 38–69 years). Histologically, the tissues were described as invasive lobular carcinoma (ILC) or invasive ductal carcinoma (IDC). BRCA resection pathology reports included the primary tumor (pT) category and pathologic lymph node status (pN) category. The tumor stagings were based on clinical tumor node metastasis (TNM). 

Several UPS components were found to be overexpressed in BRCA. The heatmaps (Figure 1B–D) highlight the upregulation of the mRNA levels of identified UPS components, including E1: UBA1 (Ubiquitin-like modifier activating enzyme 1), E2: UBE2C (Ubiquitin Conjugating Enzyme E2C), UBE2T (Ubiquitin Conjugating Enzyme E2T), and UBE2O (Ubiquitin Conjugating Enzyme E2O), and F-box factors: FBXO43 (F-Box Protein 43), FBXO15 (F-Box Protein), CCNF, and FBXL8. Scatter plots show quantitative differences in the transcript levels of E1 (Figure 1E), E2 (Figure 1F), and F-boxes (Figure 1G) in individual clinical tissues. Compared with normal tissues, the mRNA levels of these UPS components in carcinoma tissues were significantly upregulated. Our data highlighted the upregulation of E1 enzymes (Figure 1E): UBA1 by up to 3.8-fold; E2 enzymes (Figure 1F): UBE2C by up to 59.2-fold, UBE2T by up to 3.5 fold, UBE2O by up to 3.6-fold, and F-boxes (Figure 1G): FBXO43 by up to 600-fold, FBXO15 by up to 7.7-fold, and CCNF by up to 5.2-fold (*, *p* < 0.05; **, *p* < 0.01; ***, *p* < 0.001). Interestingly, among all 71 F-box family members, there are four F-boxes (FBXO43, FBXO15, FBXL8, and CCNF) with significantly upregulated mRNA levels in breast carcinoma tissues. Both FBXL8 and FBXO43 are recently reported to be pro-tumorigenic factors in BRCA [15]. However, studies on FBXO15 and CCNF in the disease are still lacking. Furthermore, it would be pertinent to investigate the E3 ligase activity to confirm the intrinsic activities of these four E3 SCF ^F-boxes^. To overcome potential bias in RNA-Seq analysis due to the small sample size, we next performed large-scale TCGA analysis by transcriptome profiling and immunohistochemical (IHC) analysis for translational determination, which also enabled us to investigate the pathophysiological relevance of the above-identified UPS components. 

### 2.2. CCNF Uniquely Shows High Expression of mRNA, but Low Protein Level in Primary BRCA 

To affirm the RNA-Seq data (Figure 1) and to further understand the correlation between the identified UPS components and the clinical status of BRCA patients, we next analyzed the TCGA (The Cancer Genome Atlas) database and performed IHC staining for selected candidates to retrospectively investigate BRCA tissues in large scale (Figure 2) and to correlate with the RNA-Seq data. For the analysis of TCGA database, a total of *n* = 1190 (with 1077 breast carcinoma tissues and 113 normal tissue counterparts) of mRNA profiles of human primary samples were determined. Consistently, the RNA-Seq database showed that the mRNA expression of E1: UBA1 (Figure 2A); E2: UBE2C, UBE2T, and UBE2O (Figure 2B); and F-boxes: CCNF, FBXL8, FBXO43, and FBXO15 (Figure 2C) were significantly upregulated in BRCA tissues (*, *p* < 0.05; **, *p* < 0.01). Box plot suggested that the overexpression of FBXL8, FBXO43, and FBXO15 may be responsible for promoting BRCA. The profile of expression of UBA1, UBE2C, UBE2T, UBE2O, and CCNF in BRCA seemed to support their involvement in cancer initiation and progression. Further studies are needed to affirm these hypotheses.

F-boxes representing key components of SCF E3 ligase are responsible for the specificity of the substrate target recognition of the ubiquitin machinery. To further understand the pathophysiological role of SCF E3 ubiquitin ligase in BRCA, we next characterized the protein profiles of FBXO43, FBXO15, and CCNF in BRCA tissues to compare with that of FBXL8 [12]. IHC was used to indicate the protein profiles in BRCA tissues (*n* = 60, with 30 breast carcinoma tissues and 30 normal tissue counterparts) (Figure 2D). Appendix A shows the clinicopathological parameters of the BRCA patients from whom breast tissue samples were obtained for IHC analysis. Interestingly, we found that although the CCNF mRNA level was elevated, its protein level remained low or even reduced in BRCA tissues compared to normal tissues (Figure 2E; ***, *p* < 0.001). TCGA analysis of malignant BRCA tissues corroboratively showed that CCNF mRNA was upregulated by 3.8-fold. Overall, both the RNA-Seq and TCGA database demonstrated high levels of CCNF mRNA in BRCA tissues, whereas its protein level was reduced by 4.5-fold. Concordantly, CCNF was recently reported to be involved in the modification of UPS [13,16]. However, the underlying mechanisms of action of CCNF in breast carcinoma remain unclear and under-explored. 

Relative to the profile of CCNF, the other F-box members displayed consistent trends of mRNA and protein expression. Similar to their mRNA expression, the protein profiles of both FBXO43 (Figure 2F) and FBXO15 (Figure 2G) were consistently strongly upregulated throughout the BRCA stagings. The FBXO43 and FBXO15 proteins were increased by 3.9–5.9 fold and 3.6–5.2 fold, respectively, in stages 0–IV, compared to normal tissues (***, *p* < 0.001). This prompted us to investigate whether and how the differential regulation of transcription and/or post-transcription expression of the F-boxes may be associated with the survival/progression of breast cancer. To this end, we performed a series of bio-functional assays for FBXO43, FBXO15, and CCNF. 

### 2.3. The Opposing Roles of F-Boxes: CCNF Suppresses while FBXO43 Promotes BRCA 

There are ≈600 E3 members from which our work on the breast cancer tissues has narrowed down to four E3 members of interest; these four members belong to the SCF^F-boxes^ subfamily of E3 ligases. These E3 SCF^F-box^ ligases are the signature candidates, which we further characterized to reveal their functional interactions in BRCA. To better understand the pathophysiological functions of the identified F-boxes (FBXO43, FBXO15, and CCNF) in BRCA, we performed specific-RNAi knockdown in breast carcinoma cells to assess the potential loss/gain-of-functions. The endogenous levels of the F-boxes were examined in two BRCA cell lines (MDA-MB231 and MCF7) and a non-cancer control cell line, MCF10A (Appendix A). Consistently, we observed an upregulation of the mRNAs of FBXO43, FBXO15, and CCNF in both MCF7 and MDA-MB231, compared to MCF10A (***, *p* < 0.001). Then, we performed RNAi knockdown of FBXO43, FBXO15, and CCNF in BRCA cells and examined the corresponding functional outcome. At a knockdown efficacy of 94% for FBXO43 (Appendix A), the cell viability after 48 h was significantly reduced by 48.5% (an average taken from MTT and CTB assays) (Figure 3A) (*p* < 0.001). Both Alamar blue and Trypan blue exclusion (TBE) approaches were used to examine cell proliferation. Compared to control siRNA, FBXO43-specific RNAi effectively caused up to 37.6% reduction in cell proliferation of both MCF7 and MDA-MB231 cells (Figure 3B, *p* < 0.001). A recent study showed that highly expressed FBXO43 is associated with a shortened duration of disease-free survival and overall survival in BRCA patients [15]. In agreement with this report, our functional studies here showed that FBXO43 plays a critical oncogenic role in BRCA. In comparison, although the knockdown efficiency of 89% was achieved (Appendix A), FBXO15 did not affect the cell viability and cell proliferation, implying that FBXO15 probably acts independently of FBXO43 and CCNF. Nevertheless, studies would be required to investigate the pathophysiological role of FBXO15 and its underlying mechanisms potentially involved in the disease. 

Similar to FBXO15, we found that CCNF knockdown did not affect either cell survival or cell proliferation. CCNF was proposed to play a cancer-suppressive role in hepatocellular carcinoma [17]. Prompted by our observation of low protein level in primary BRCA tissues (Figure 2D), we further performed CCNF overexpression in BRCA cell lines (Appendix A) to investigate its functional roles. Interestingly, we found that CCNF overexpression resulted in a significant reduction (48.5%) in cell viability after 48 hours (Figure 3E) (*p* < 0.001). CCNF overexpression also effectively reduced cell proliferation by up to 51.3% in both MCF7 and MDA-MB231 cells (Figure 3F, *p* < 0.001). Figure 3G shows that CCNF overexpression significantly increased early apoptosis of MCF7 and MDA-MB231 cells (indicated by Annexin V^+^/7AAD^−^) by 24% and 20%, respectively. Representative histograms of apoptosis assays are shown in Appendix A (*p* < 0.001). Further investigation of caspase activities supported that both caspases-9 and -3 were activated in CCNF-overexpressed cells (Figure 3H), indicating that CCNF suppresses BRCA cell through the intrinsic apoptosis pathway. 

### 2.4. Overexpression of CCNF Effectively Suppressed BRCA Migration and Invasion

To ask whether CCNF plays a role in BRCA metastasis, we studied the effects of CCNF overexpression in cell migration and invasion assays in BRCA cells. For cell migration assay, we monitored over 30 h time points for both MCF7 (less invasive) and MDA-MB231 cells (highly invasive), and we showed a significant reduction of cell migration by 30% and 55%, respectively, when these cells overexpressed CCNF compared to control pcDNA3.1 vector alone, *p* < 0.001 (Figure 4A–D). Concordantly, the overexpression of CCNF dramatically inhibited BRCA invasion by >3-fold (compared to pcDNA3.1 controls), with remnants of only 26.7% and 35.6% invasive cells in MCF7 and MDA-MB231 cells, respectively (*p* < 0.001) (Figure 4E, Appendix A). These results suggest that CCNF is an efficient suppressor of metastatic potential of BRCA. It should be noted that invasion assays over 24 h such as this could be influenced by simultaneous cell proliferation. However, we observed the doubling time for MCF7 and MDA-MB231 cells to be 26 h and 23 h, respectively (Figure 3B,D,F), which cannot contribute to the clear effects of CCNF overexpression against the migratory/invasion behavior of the BRCA cells. Therefore, CCNF suppresses BRCA metastatic potential. Being a key component of E3 ligase, it was pertinent for us to further investigate CCNF-associated ubiquitin machinery involved in BRCA development.

### 2.5. Knockdown of FZR1 and FBXL8 Upregulated CCNF, but Overexpression of CCNF Suppressed RRM2 in BRCA

As an anti-tumorigenic factor, CCNF is under tight control within the ubiquitin machinery [13]. However, the mechanisms underlying CCNF regulation remain unclear in breast carcinogenesis. Our in silico modeling prediction revealed that CCNF interacts with other E3 SCF^F-boxes^ such as RBX1, CUL1, CDC20, SKP1, FZR1, FBXL8, CCNB1, and FBXW7 (Figure 5A–C). These results highlight the potential UPS candidates associated with CCNF activity, particularly the E3 ligase itself or the co-regulator of E3 ligase, e.g., FZR1 [18,19,20] and/or FBXL8 [12], which are known to promote BRCA. Since CCNF was recently identified as a direct substrate of the FZR1 ubiquitin ligase [21], we focused our attention on FZR1. Furthermore, other studies reported that CCNF (an E3 ligase) targets RRM2 (Ribonucleotide reductase M2), a pro-tumorigenic protein, which is linked to poor survival rate in patients with skin melanoma and associated with its protein degradation activity [22]. Concordantly, previous studies showed that the expression of RRM2 (an oncogenic factor) is elevated in breast cancer tissues and cells [23,24]. Here, we further investigated the RRM2 regulatory mechanism in BRCA. *Gilt by association*, we perceived the potential significance of investigating whether pro-tumorigenic FBXL8, FZR1, and RRM2 are collaboratively responsible for CCNF-associated networks. This was with a view to providing explanations on how CCNF-associated ubiquitin machinery impacts BRCA progression. We found that knockdown of either FBXL8 or FZR1 caused the accumulation of CCNF proteins in MCF-7 and MDA-MB231 cells (Figure 5D, upper panel). Double-knockdown of both the FBXL8 and FZR1 further increased CCNF protein levels (red boxes), suggesting that the FBXL8 (an E3 SCF^F-box^ ligase) and FZR1 (anaphase promoting complex/cyclosome (APC/C) E3 co-regulator) could be responsible for CCNF protein degradation. On the other hand, CCNF overexpression consistently reduced the RRM2 protein level in both of the BRCA cell lines (Figure 5D, lower panel). These results highlight the possibilities that on one hand, CCNF is itself modulated via protein degradation by the E3 ligases (FBXL8) or co-regulator (FZR1), but on the other hand, CCNF also exerts an E3 ligase function, which presumably degrades the RRM2 protein in BRCA. 

Co-immunoprecipitation (Co-IP) further showed that both FZR1 and FBXL8 pulled down CCNF in the BRCA cells (Figure 5E, upper and middle), indicating that CCNF could be a conserved substrate specifically targeted by FZR1 and FBXL8 E3 ligases. IP with control IgG or IgG_2b_ demonstrated the specificity of the assay. Furthermore, experimental downregulation of CCNF has been shown to increase RRM2 expression, leading to the risk of carcinogenesis [22]. We showed that CCNF itself pulled down RRM2 (Figure 5E, lower), further corroborating that CCNF is an E3 ligase that specifically targets RRM2 protein and presumably caused its degradation. Concordantly, RRM2 is recently demonstrated as a pro-tumorigenic factor in BRCA [23]. Thus, CCNF-triggered RRM2 reduction (in the CCNF-RRM2 axis) likely unleashes anti-cancer mechanisms against BRCA. Figure 5F summarizes the functional network of E3 SCF^F-box^ ligases (FBXL8, CCNF), E3 co-regulator (FZR1), and RRM2 (substrate of E3 SCF^F-box^ CCNF) in BRCA.

## 3. Discussion

UPS has been recognized as a pivotal contributor to cancer development and progression, including BRCA [12,25]. Identifying the critical UPS molecules selectively activated in BRCA patients would be useful in the development of targeted therapy for BRCA. For the first time, we characterized the global transcriptome profiles of BRCA primary tissues and identified several crucial UPS components which are transcriptionally upregulated, including E1: UBA1, E2: UBE2C, UBE2T, and UBE2O and F-box factors (key members of SCF E3 ligases): FBXO43, FBXO15, CCNF, and FBXL8. In the future, these identified molecules can be developed as either sole or combinatorial anti-cancer drugs for the treatment of heterogenous cancer subtypes.

CCNF was reported to be significantly repressed in hepatocellular carcinoma tissues [17]. Interestingly, we observed that compared to FBXO43, FBXO15, and FBXL8, the CCNF mRNA was uniquely high, while its protein level was low in primary BRCA tissues. Concordantly, we showed that unlike FBXO43 and FBXL8, which promote BRCA tumorigenesis (Figure 3A,B and [12]), CCNF effectively suppressed cancer progression (Figure 3E–H, Figure 4). CCNF overexpression significantly inhibited cell growth, proliferation, and metastatic potentials in BRCA. At least two E3 ligases, FBXL8 and FZR1, were found to specifically target CCNF protein. FBXL8 and FZR1 are potentially responsible for the degradation of CCNF, which promotes BRCA, since knockdown of FBXL8 and/or FZR1 caused CCNF protein to accumulate (Figure 5D,E). Thus, we propose that CCNF plays a cancer suppressor role, but its protein level is downmodulated in the disease, probably via FBXL8/FZR1-mediated degradation. In support of our findings, future studies would be warranted to explore whether CCNF is commonly downregulated by ubiquitination in other cancers.

Our in silico prediction indicated probable E3 binding partners of CCNF (Figure 5A–C). In vitro studies showed that either FBXL8 or FZR1 consistently pulled down CCNF. Congruently, the knockdown of FBXL8 or FZR1 caused an elevation in the CCNF protein level, suggesting the potential reciprocal functional relationship in CCNF- and FBXL8- (or FZR1-) dependent protein degradation. Since at least two—E3 ligase and E3 co-regulator—are found to be responsible for CCNF protein degradation, it is also possible that distinct E2–E3 combinations could coordinate to ubiquitinate a common substrate [26], such as CCNF, in this case. It is plausible that besides E3, different E2s in the UPS also respond to the pathophysiological conditions to modulate the CCNF-FBXL8 (or CCNF-FZR1) axis, which could exacerbate the tumorigenic and metastatic progression. 

*Gilt by association*, RRM2 appears to be a substrate of CCNF in normal healthy breast tissues. RRM2 is recently reported to be a pro-metastatic factor in BRCA [23]. Concordant with this report, we observed that CCNF overexpression dramatically reduced the RRM2 protein level, suggesting that the RRM2 protein level could be downmodulated by CCNF-associated protein degradation (Figure 5D,E). Taken together, we propose that in healthy individuals, CCNF acts as a functional E3 ligase to effectively reduce RRM2 protein level, hence inhibiting breast carcinogenesis (Figure 6). On the other hand, in BRCA patients, CCNF is suppressed by dominant counterpart E3 ligases, FBXL8 and FZR1. CCNF is known to be strictly controlled by UPS networks (Figure 6, green boxes [13,21,22,27,28,29,30,31,32,33,34,35,36]). The reduction of cancer suppressor, CCNF, causes uncontrolled cell survival and proliferation, leading to breast carcinogenesis (Figure 6). Therefore, in the future, it would be interesting to design anti-BRCA strategies by counterbalancing the activities of E2 and E3 ligases, for example, to (1) maintain CCNF to suppress tumor-promoter RRM2 and/or (1) inhibit FBXL8 and FZR1 to reduce their targeting of tumor-suppressor CCNF protein. Altogether, we propose that CCNF-associated ubiquitin machinery fine-tunes the regulation of BRCA development. Investigations on the relevant signaling molecules driven by CCNF will aid BRCA prediction, early diagnosis, and the development of new BRCA therapies.

## 4. Materials and Methods

### 4.1. Tissue Samples

The primary samples used in this study were prepared as described previously [12]. Briefly, in total, *n* = 70 (35 tumor tissues and 35 normal tissues) from breast cancer patients were acquired from the BioBank, Taipei Medical University, Taiwan. Experiments were performed in accordance with institutional guidelines (Taipei Medical University-Joint Institutional Review Board; IRB: N201512055 and N201803107). 

### 4.2. Cell Lines and Reagents 

Human breast cancer cell lines (MCF7, MDA-MB231) and a non-carcinoma breast cell line control (MCF10A) were obtained from American Type Culture Collection (ATCC). The MCF7 and MDA-MB231 were cultured in complete DMEM medium (Gibco, Waltham, MA, USA). MCF10A was cultured in complete DMEM/F12 medium (Gibco). Both of the complete media were supplemented with 10% FBS (Thermo Scientific, Waltham, MA, USA), 100 U/mL penicillin, and 100 μg/mL streptomycin (Invitrogen, Carlsbad, CA, USA). Antibodies used in IHC analysis were anti-CCNF antibody (PA5-99463, Invitrogen), anti-FBXO43 (PA5-21622, Invitrogen), and anti-FBXO15 antibody (ABIN1386824, antibodies-online GmbH). 

### 4.3. NGS (Next-Generation Sequencing)-Based RNA-Seq (Sequence) Analysis

The primary BRCA tissue samples were prepared as previously described [12]. Details are in the Appendix A, including (1) RNA isolation for NGS-based RNA-Seq., (2) mRNA library preparation and NGS-based RNA-Seq., and (3) RNA Sequence analysis. 

### 4.4. Quantitative Real-Time PCR (qRT-PCR)

qRT-PCR was performed as previously described [12]. Details are in the Appendix A. The primers were human FBXO43 (207 bp product): sense, 5’-CTAGCAATCCTCCTGCCTTG-3’; antisense, 5’-TGTTTGGAGAGATGGGAAGG-3’; human FBXO15 (231 bp product): sense, 5’-TCAGACTTGCCCACACTGAG-3’, antisense, 5’-ACGTTGGAAGTCACCACTCC-3’; human CCNF (167 bp product): sense, 5’-AATGGCAGTGGAAACTTTGG-3’; antisense, 5’-AGTGGGAGTGTGGAAACAGG-3’; human GAPDH (131 bp product): sense, 5’-GTCTCCTCTGACTTCAACAGCG-3’, antisense, 5’-ACCACCCTGTTGCTGTAGCCAA-3’. All expression values were normalized based on GAPDH as an endogenous control.

### 4.5. RNAi Knockdown and Overexpression of CCNF in BRCA Cells

FBXO43 siRNA (siGENOME Human FBXO43-SMARTpool, M-025904-02-0005), FBXO15 siRNA (siGENOME Human FBXO15-SMARTpool, M-016503-00-0005), CCNF siRNA (siGENOME Human CCNF-SMARTpool, M-003215-02-0005), and control (scrambled) siRNA were purchased from Dharmacon (Lafayette, CO, USA) and Invitrogen, respectively. The transfection experiments were performed as previously described [12]. Details are in the Appendix A. Briefly, siRNA was transfected into MCF7 or MDA-MB231 cells (at 5 × 10^5^ cells per well of a 6-well plate) using 7.5 μL TransIT-X2 Transfection Reagent (Mirus, Madison, WI, USA) with 25 nM of siRNAs per well. The efficiency of RNAi knockdown was determined by real-time PCR of FBXO43, FBXO15, and CCNF mRNAs (Appendix A). CCNF–pcDNA3.1 (Accession No: NM_001761.3) and pcDNA3.1 vector alone plasmids were used for CCNF overexpression study (purchased from GenScript Biotech, Singapore, Singapore). For BRCA cells, DNA transfection was conducted with 8 × 10^5^ cells per well of a 6-well plate using 4 ml X-tremeGENE Transfection Reagent (Roche, St. Louis, MO, USA) with 2 mg DNA plasmids [3]. The transfection efficiency for cDNA overexpression was determined by real-time PCR of CCNF (Appendix A). 

### 4.6. Cell Viability Assay

Cell viability studies were performed as previously described [12]. Details are in the Appendix A. To measure the BRCA cell survival in response to the presence or absence of FBXO43 (or FBXO15 or CCNF), both MCF7 and MDA-MB231 cells were treated with either (1) specific siRNA (targeting to FBXO43, FBXO15, or CCNF), control siRNA, or PBS, or (2) CCNF–pcDNA3.1, control pcDNA3.1, or PBS for 16 h. 

### 4.7. Cell Proliferation Assay

Cell proliferation was studied as previously described [12]. Details are in the Appendix A. Briefly, cells were transfected with target-specific siRNA, control scrambled siRNA, CCNF–pcDNA3.1, or control pcDNA3.1 vector alone for 16 h. Then, Alamar blue was used to measure cell growth. 

### 4.8. Cellular Apoptosis Assay

Cell apoptosis was measured as previously described [12]. Details are in the Appendix A. Briefly, CCNF–pcDNA3.1 plasmid was transfected into MCF7 and MDA-MB231 cells; the control was empty vector, and the background control treatment used PBS. After 24 h, the cells were examined for apoptosis. 

### 4.9. Caspase-9 and -3 Assays

Caspase-9 and -3 were measured as previously described [12]. Details are in the Appendix A. 

### 4.10. Cell Migration and Invasion Assays

Cell migration and invasion studies were performed as previously described [12]. Details are in the Appendix A. Cell migration assay was carried out 24 h after transfection of the BRCA cells with CCNF–pcDNA3.1 or controls (empty pcDNA3.1 vector and PBS). For cell invasion assay, 24 h after treatment with CCNF–pcDNA3.1 (or pcDNA3.1 or PBS control), BRCA cells were plated onto the chambers.

### 4.11. Statistical Analysis

Data were expressed as means ± SD from three independent experiments, with three replicates per sample/condition tested. Differences between averages were analyzed by two-tailed Student’s t-test. Significance was set at *p*-value of <0.01 (**, *p* < 0.01; ***, *p* < 0.001). The acquired data from FACS were analyzed with BD FACSuite^TM^ software (BD Biosciences, San Jose, CA, USA). The relative migration rate indicated as percentage of gap closure was calculated using Image J analysis software. All target signals from IHC were quantified by HistoQuest software (TissueGnostics, Vienna, Austria). 

## 5. Conclusions

Altogether, we showed a signature of CCNF-associated E3 ligases (or E3 co-regulator) which collaboratively or reciprocally modulate CCNF-associated ubiquitin machinery in breast cancer. Our findings bear translational potential to target BRCA through modulation of the partnership axes of: (i) CCNF-FBXL8, (ii) CCNF-FZR1, (iii) CCNF-RRM2. 

## Figures and Tables

**Figure 1 cancers-13-02873-f001:**
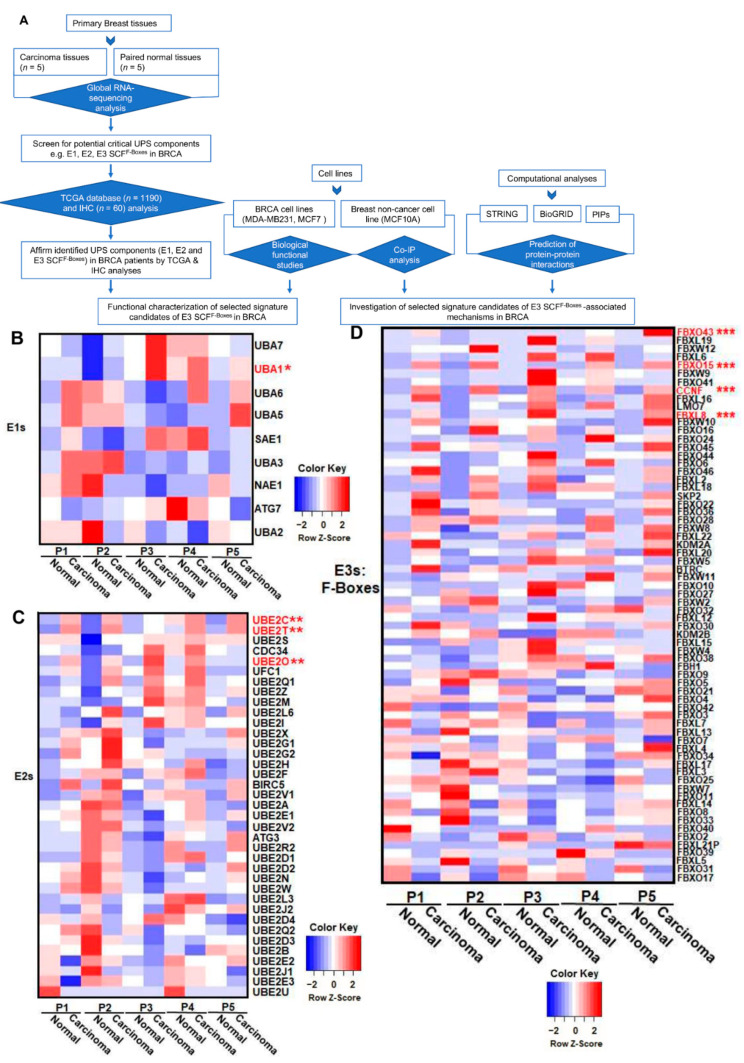
RNA-Seq analysis revealed the transcriptional profiles of the key components of SCF E3 ligase in human breast carcinoma tissues. (**A**) Experimental plan and strategic workflow in this work. To characterize the expression profiles in human breast cancer (BRCA), we performed a retrospective study by Next-Generation Sequencing-based RNA-Sequence (NGS-based RNA-Seq) approach. Five breast carcinoma tissues and the corresponding normal counterparts were characterized for their global mRNA profiles. The profiles of (**B**) E1, (**C**) E2, and (**D**) F-box family members are shown by heatmap analysis. The identified significant targets are highlighted in red, including E1: UBA1; E2: UBE2C, UBE2T, UBE2O; and F-box: FBXO43, FBXO15, CCNF, FBXL8. In the heatmap, blue color represents low expression, while red represents high expression. (**E**), (**F**), and (**G**) show quantitative mRNA expression of the identified E1, E2, and F-box components, respectively, from each breast cancer patient. FBXL8 was reported recently [12]. * the corresponding clinicopathological information for IHC is shown in Appendix A [12]. Each data point is generated by the R package, DESeq2, with default log fold-change thresholds of −2 and 2 and an adjusted *P*-value threshold of 0.05. (*, *p* < 0.05; **, *p* < 0.01; ***, *p* < 0.001). * The relevant data /resources e.g., pathology data from patients used for RNA-seq analysis and IHC staining, have been referred to in our previous publication (“Human FBXL8 Is a Novel E3 Ligase Which Promotes BRCA Metastasis by Stimulating Pro-Tumorigenic Cytokines and Inhibiting Tumor Suppressors”, Cancers 2020, 12, 2210; doi:10.3390/cancers12082210—[12]). For ease of reference, we have provided a brief outline above (*).

**Figure 2 cancers-13-02873-f002:**
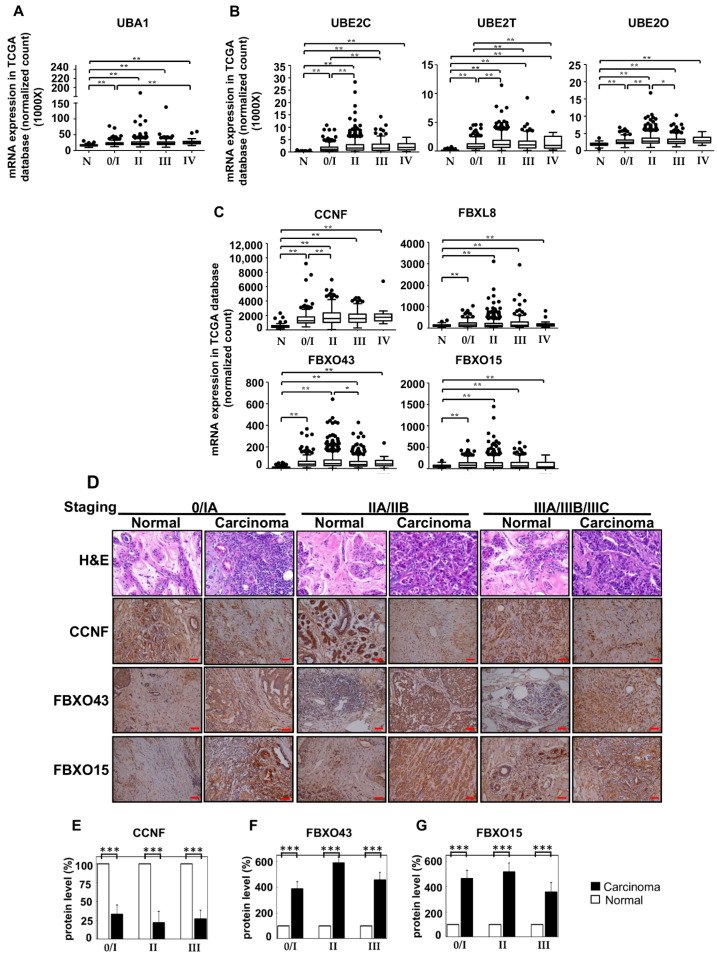
Transcriptional profiles of the main components of SCF E3 from the TCGA database of human primary BRCA tissues. To clarify the potential of the identified SCF E3 ligase, both TCGA database results (**A**–**C**) and IHC results (**D**) were analyzed. (**A**) As a key E1 component, UBA1 was elevated in BRCA tissues throughout all disease stages. (**B**) UBE2C, UBE2T, and UBE2O were identified as the main E2 components, showing significant overexpression in breast carcinoma tissues, particularly in advanced stages. (**C**) Investigation of FBXL8 profiling was recently studied in BRCA tissues [12]. Here, we further reported that CCNF, FBXO43, and FBXO15 were identified as the main E3 F-box components, which show significant overexpression in carcinoma tissues. Y-axis: normalized count, X-axis: pathological cancer stages. The number of samples in each stage is shown in parentheses: stage I/0 (*n* = 182), stage II (*n* = 624), stage III (*n* = 251), and stage IV (*n* = 20). N: normal (*n* = 113). *, *p* < 0.05; **, *p* < 0.01. (**D**) shows H&E and immunofluorescence staining. Protein expression levels of CCNF, FBXO43, and FBXO15 were examined in both breast carcinoma (*n* = 30) and paired normal breast tissues (*n* = 30). The corresponding quantitative results of CCNF, FBXO43, and FBXO15 in IHC are shown in (**E**–**G**), respectively. Consistent with our observation of the mRNA levels, the FBXL8 [12], FBXO43, and FBXO15 proteins are significantly upregulated in breast carcinoma tissues compared with normal tissues. However, the opposite trend of CCNF was observed in BRCA tissues; CCNF mRNA was elevated, but its protein level was significantly reduced, indicating post-transcriptional modifications and degradation of CCNF protein. The corresponding clinicopathological information for IHC are shown in Appendix A [12]. ImageJ software was used to analyze IHC staining intensity (total >300 sections/each target protein). Brown color indicates DAB dye, which stains the protein of interest. Scale bar is 100 μm, shown as the red color line (—). ***, *p* < 0.001.

**Figure 3 cancers-13-02873-f003:**
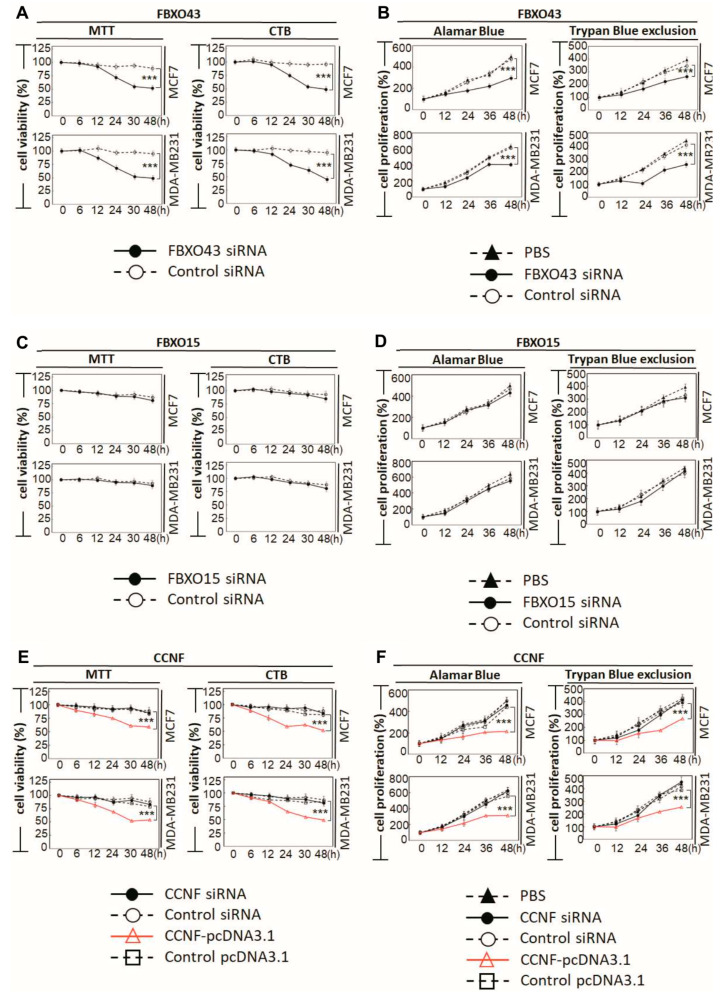
CCNF is a cancer suppressor, but FBXO43 is a cancer promoter in BRCA. To elucidate the functional roles of FBXO43, FBXO15, and CCNF in breast carcinoma, we analyzed two human BRCA cell lines: MCF-7 and MDA-MB231, compared to a non-cancer control breast cell line, MCF10A. qRT-PCR shows the endogenous mRNA levels of FBXO43 (Appendix A), FBXO15 (Appendix A), and CCNF (Appendix A) in BRCA cells, compared to MCF10A cells. The mRNA levels of FBXO43, FBXO15, and CCNF are significantly upregulated in breast carcinoma cells. The viability of BRCA cell lines are shown by MTT and CTB assays. For each time point, cell counts were normalized to the corresponding PBS control. Alamar blue or Trypan blue exclusion tests were used to examine cell proliferation. (**A**,**B**) Knockdown of FBXO43 significantly reduced cell viability (**A**) and cell proliferation (**B**) in BRCA. (**C**,**D**) Knockdown of FBXO15 showed no effect on BRCA cell viability (**C**) and proliferation (**D**). (**E**,**F**) CCNF overexpression (but not RNAi) significantly suppressed BRCA cell survival (**E**) and proliferation (**F**) (red lines). (**G**) Annexin V and 7-aminoactinomycin (7-AAD) double staining was conducted to examine early apoptosis (AnnexinV^+^/7AAD^—^). (**H**) Overexpression of CCNF in MCF7 and MDA-MB231 cells resulted in apoptosis via activation of caspases-9 and -3, suggesting that high levels of CCNF suppresses BRCA advancement through cell suicide. qRT-PCR analysis was used to measure the efficacy of RNAi knockdown (targeting FBXO43, FBXO15, or CCNF) or CCNF overexpression (Appendix A). Optimized transfection efficacy with FBXO43 (or FBXO15 or CCNF)-specific siRNA is shown in both MCF7 and MBA-MD231 cells compared to controls using PBS and scrambled siRNA (***, *p* < 0.001). CCNF overexpression was optimal at 48 h, where up to a 12–15 fold increase of CCNF mRNA was achieved in both cell lines. Representative histograms of apoptosis assay are shown in Appendix A. Data are means ±SD (*n* = 3). ***, *p* < 0.001.

**Figure 4 cancers-13-02873-f004:**
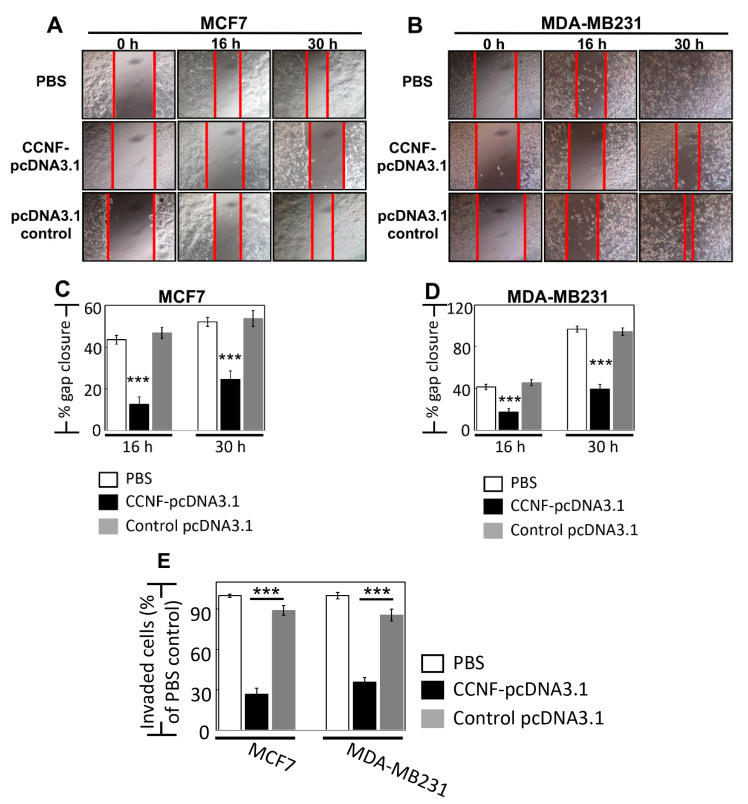
Overexpression of CCNF effectively suppressed BRCA migration and invasion. Cell migration assay was performed for (**A**) MCF7 and (**B**) MDA-MB231 cells. (**C**) and (**D**) show the corresponding quantitative results. Compared with control pcDNA3.1 vector alone, treatment of CCNF–pcDNA3.1 significantly reduced BRCA cell migration. For each treatment, the migration rate (% gap closure) was normalized to the corresponding 0 h time point controls. (**E**) Cell invasion is quantified. The representative microscopy images are shown in Appendix A. We showed that CCNF overexpression effectively reduced cell invasion. Data are representative of means ± SD (*n* = 3). ***, *p* < 0.001.

**Figure 5 cancers-13-02873-f005:**
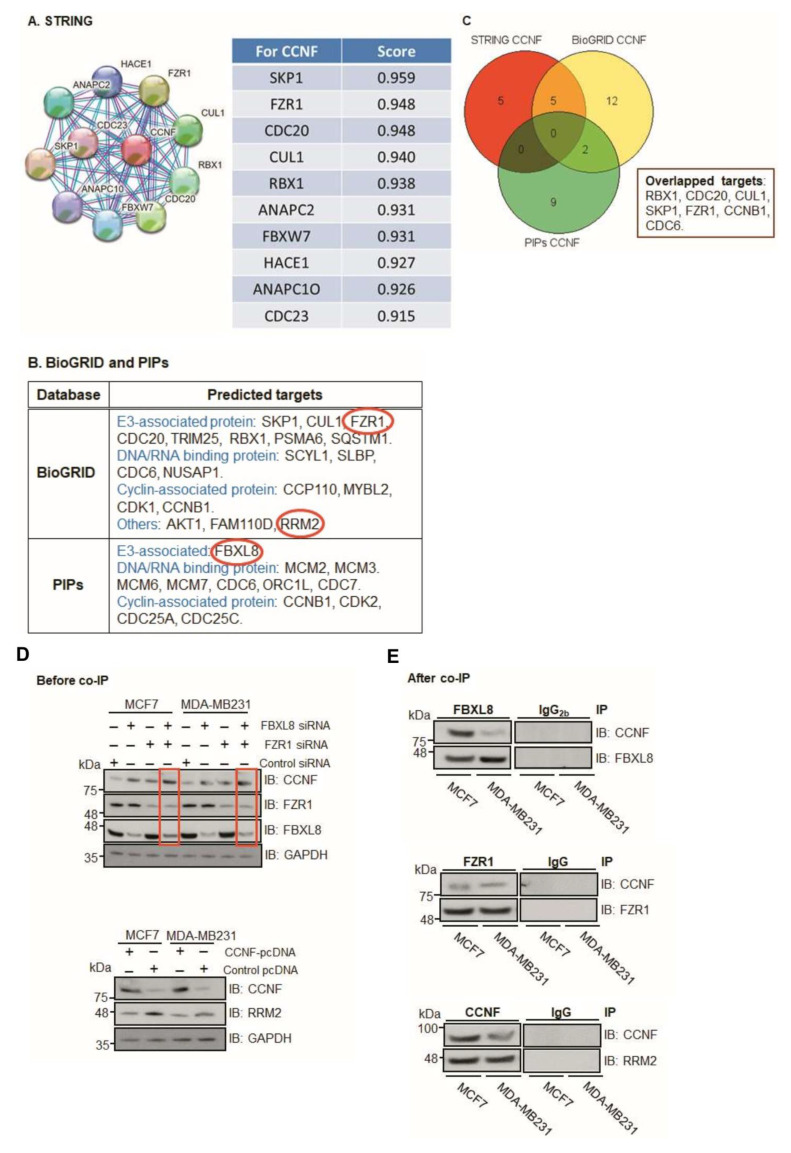
Computational modeling prediction of CCNF-associated proteins. To search for potential proteins associated with CCNF, three databases were used including STRING (https://stringdb.org/, accessed on 1 May 2020), BioGRID (https://thebiogrid.org/, accessed on 10 April 2020), and PIPs (http://www.compbio.dundee.ac.uk/wwwpips/, accessed on 31 Mar 2020). (**A**) The predicted factors shown by STRING may be involved in CCNF binding. (**B**) A summarized list of CCNF-binding proteins based on BioGRID and PIPs. (**C**) Protein targets including RBX1, CDC20, CUL1, SKP1, FZR1, CCNB1, and CDC6, which were predicted by any two databases or in all three databases. (**D**) To confirm whether CCNF is under the regulation of FBXL8 and/or FZR1-dependent protein degradation, RNAi of FBXL8 or FZR1 was performed in MCF7 and MDA-MB231 cells (upper panel). The immunoblotting results show that double knockdown increased the level of CCNF (red boxes). GAPDH was used as a loading control for immunoblotting. To confirm whether RRM2 was regulated by CCNF-dependent protein degradation, CCNF overexpression was performed in MCF7 and MDA-MB231 cells, followed by immunoblotting analysis; results (lower panel) show a decrease in RRM2 when CCNF was overexpressed. (**E**) Co-IP of FBXL8 (upper) or FZR1 (middle) or CCNF (lower) or control anti-IgG antibodies in BRCA cells followed by immunoblottings of total cell lysates with anti-FBXL8, anti-FZR1, anti-CCNF, or anti-RRM2 antibodies showed that both FBXL8 and FZR1 pull downed CCNF. Furthermore, CCNF pulled down RRM2 in BRCA cells. Figure 5F shows the functional network of signature E3 ligases (E3 SCF^F-boxes^). FBXL8 and FBXO43 are pro-tumorigenic E3 ligases, while CCNF is an anti-tumorigenic E3 ligase. FZR1 is an APC/C E3 co-regulator that was also found to downregulate CCNF. RRM2 is an oncogenic factor that was found to interact with CCNF, and CCNF negatively regulates RRM2 in BRCA. Further investigations using double knockdown of FZR1 and FBXL8 followed by RRM2 activity determination would be interesting to explore the axis of FZR1/FBXL8-RRM2 in the future.

**Figure 6 cancers-13-02873-f006:**
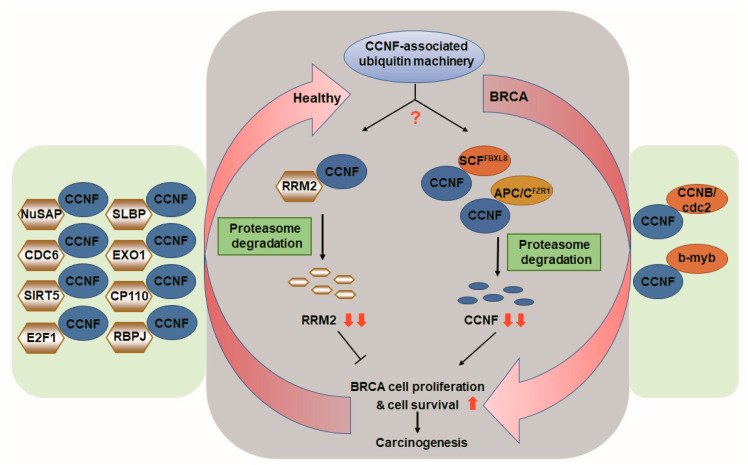
CCNF-associated ubiquitin machinery in health and in breast cancer. Based on the literature and our current findings, we propose that in healthy individuals, CCNF acts as a pivotal E3 ligase, which is responsible for protein degradation of tumorigenic RRM2 (gray background, left), hence suppressing cancer cell proliferation, survival, and carcinogenesis. In contrast, in patients with breast carcinoma, CCNF is suppressed by E3 ligase FBXL8 (SCF^FBXL8^) and E3 co-regulator FZR1 (APC/C^FZR1^), resulting in ubiquitin degradation of CCNF. The resulting low level of CCNF protein leads to increased cell survival, proliferation, and cancer formation in BRCA (gray background, right). The red arrows indicate the Yin–Yang counter-balance driven by CCNF-associated ubiquitin machinery. The binding partners of CCNF shown by previous studies [13,21,22,27,28,29,30,31,32,33,34,35,36] are highlighted in the green boxes.

## Data Availability

The data presented in this study are available in “A novel signature of CCNF-associated E3 ligases collaborate and counter each other in breast cancer” or Appendix A here.

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
