# Peer review of "A Novel Signature of CCNF-Associated E3 Ligases Collaborate and Counter Each Other in Breast Cancer"

_cancers, 2021, doi:10.3390/cancers13122873_

Round 1
Reviewer 1 Report
In this paper, Chang et al. described FBXL8, FZR1, CCNF and RRM2 network regulated Breast cancer development. However, the manuscript is hard to follow. One of the reasons is that all the data showed in this paper are quite random. According to the work model, the story is about CCNF-associated ubiquitin machinery. CCNF is regulated by FBXL8 and FZR1 and it is involved in the regulation of RRM2. However, in most of the key experiments, Figure 1, RNA-seq, Figure 2, TCGA data and IHC data and Figure 3, cell viability and cell proliferation experiments, no data (rare) is about FZR1 or RRM2 or FBXL8. Thereby, it is hard to predict whether the work model is correct or not.
To be more specific:
- In patient samples, CCNF and FBXL8 mRNA were upregulated. In the TCGA data, the result is consistent. However, protein level of CCNF in breast tissue was reduced. Do you think it is because FBXL8 trigger the degradation of CCNF? Do you stain RRM2 in those tissues? Do you expect to see high level of RRM2 according to your work model? You compared CCNF mRNA level in MDA-MB231, mcf7 to MCF10A, do you check CCNF protein level in those cell lines? When overexpression of CCNF, do you have western to show how much it is overexpressed? Sometime, high expression level of ectopic protein can be toxic too. Do you check RRM2 expression level? If you double knockdown FZR1 and FBXL8, do you see RRM2 expression level change? Will knockdown RRM2 or overexpression RRM2 change cell viability and cell proliferation?
- Overexpression of CCNF effectively suppressed BRCA migration and invasion. Do you think FBXL8 promote Breast cancer metastasis by inhibiting CCNF? Do you think E3 function of FBXL8 directly involved in this process?
Reviewer 2 Report
The authors present an interesting and well-conducted study. I have several questions for clarification of the findings.
The use of BRCA in the title, without previously defining it, is confusing. The reader's immediate impression is that the subjects in the study are BRCA mutation carriers. This is reinforced in the manuscript by referring to factors as being a BRCA promoter. Breast cancer should be spelled out in the title, and it made clear that this is sporadic breast cancer population (if it is).
Abstract. It is stated that FBXL8 and FZR1 pulled down CCNF, which should result in increased RRM2 (since CCNF reduces RRM2) which is tumorigenic, and yet it is concluded that E3 ligases modulate CCNF anti-cancer activity. It is therefore not clear how modulation of CCNF-FBXL8 or CCNF-FZR1 could serve as a target for therapy
It would appear from the heat maps that other E1, E2 and E3 ligases are upregulated in carcinoma. How were the four E3 ligases selected for further study? It would be helpful to include a bioinformatics sections on analysis of the RNAseq data.
Line 48. It is stated that “The profile of expression of UBA1, UBE2C, UBE2T, UBE2O and CCNF in BRCA seemed to support their involvement in cancer initiation and progression.” The observations were made in breast cancer tissue compared with normal tissue, and could have resulted from malignant transformation. There is no data presented to indicate they are involved in the initiation and development of breast cancer. Please clarify this statement.
Fig 6, legend: Different labels are used for CCNF in normal tissue (SCFCCNF) vs. tumor (CCNF). What is the evidence that the ligases are destroying it in cancer and why should they do this and not in normal – are the authors suggesting there are differences in these proteins according to tissue site?
Round 2
Reviewer 1 Report
All my questions are properly addressed. I appreciate it. Thank you.
Author Response
We thank the reviewer for accepting our responses.